# Experiences and attitudes of Danish men who were sperm donors more than 10 years ago; a qualitative interview study

**Stina Lou**[1,2,3]*, **Stina Bollerup**[1], **Morten Deleuran Terkildsen**[1,3], **Stine Willum Adrian**[4], **Allan Pacey**[5], **Guido Pennings**[6], **Ida Vogel**[2,3,7], **Anne-Bine Skytte**[8]

**1** DEFACTUM–Public Health Research, Central Denmark Region, Aarhus, Denmark, **2** Department of Clinical Medicine, Center for Fetal Diagnostics, Aarhus University, Aarhus, Denmark, **3** Department of Clinical Medicine, Aarhus University, Aarhus, Denmark, **4** Department of Culture and Learning, Aalborg University, Copenhagen, Denmark, **5** Department of Oncology and Metabolism, The Medical School, The University of Sheffield, Sheffield, United Kingdom, **6** Department of Philosophy and Moral Science, Bioethics Institute Ghent (BIG), Ghent University, Gent, Belgium, **7** Department of Clinical Genetics, Aarhus University Hospital, Aarhus, Denmark, **8** Cryos International Sperm and Egg Bank, Aarhus, Denmark

* Stina.lou@rm.dk

**Data Availability Statement:** For this study, participants only consented to external data

## Abstract

### Background

More knowledge about the long-term impact of sperm donation is essential as the donor's attitude towards donation may change over time. Personal and social developments may prompt a rethinking of previous actions and decisions, or even regret. Thus, the aim of this study was to explore the experiences and attitudes of men who were sperm donors more than 10 years ago.

### Methods

From May to September 2021, semi-structured, qualitative interviews were conducted with 23 former donors (> 10 years since last donation) from Cryos International sperm bank. Two participants were non-anonymous donors and 21 were anonymous. The interviews were conducted by phone or via video (mean 24 minutes). All interviews were recorded, transcribed verbatim and rendered anonymous. Data were analyzed using thematic analysis.

### Results

The analysis showed that most men had been donors for monetary and altruistic purposes, and now considered sperm donation as a closed chapter that was 'unproblematic and in the past'. Most men valued anonymity and emphasized the non-relatedness between donor and donor conceived offspring. Knowledge about recipients and donor offspring was seen as 'damaging' as it could create unwanted feelings of relatedness and responsibility towards them. All men acknowledged donor conceived persons' potential interests in knowing about their genetic heritage in order to understand appearance and personal traits, but also emphasized the donors' rights to anonymity. Potential breach of anonymity was generally

sharing in anonymized form. Since full transcripts cannot be fully anonymized due to the highly individual context, the transcripts can not be shared in a public repository. However, data can be made available from DEFACTUM at the Central Denmark Region, Aarhus, Denmark (contact via defactum@rm.dk) for researchers who meet the criteria for access to confidential data.

**Funding:** This study was funded by Cryos International, https://www.cryosinternational.com/da-dk/dk-shop/professionel/forskning/vores-videnskabelige-forskning/ (to SL) ,and the Novo Nordisk Foundation, https://novonordiskfonden.dk/en/ (Grant No. NNF16OC18722 to IV). Cryos International assisted with recruitment of participants, but otherwise the funders had no role in study design, data collection and analysis, decision to publish, or preparation of the manuscript.

**Competing interests:** SL, SB, MDT and IV declare no conflict of interest. AP, GP and SWA are members of the Cryos External Scientific Advisory Committee. GP has received honorarium for this. ABS is an employee of Cryos International. This does not alter our adherence to PLOS ONE policies on sharing data and materials. Due to anonymity issues, data material can only be made available upon reasonable request to the first author.

considered 'highly problematic' as it was expected to disturb their families and force a relationship on them.

## Conclusion

This study reports on former donors who might not have volunteered for research due to lack of interest or protection of privacy. The majority of men valued anonymity and clearly demarcated a line between sperm donation and fatherhood, which was enforced by not knowing about the donor offspring or recipients.

## Introduction

Men's reasons for becoming a sperm donor have been extensively researched, pointing to mainly financial and altruistic incentives [1,2]. However, less research concerns the long-term impact of having been a sperm donor, and this research tends to focus on the perspectives of donors who are willing to be identified. Studies have investigated donors' motivations for disclosing their identity, e.g., in DNA databases [3,4], and their attitudes towards and experiences of contact with donor conceived offspring [5,6]. However, in order to adequately counsel prospective sperm donors about the potential long-term impact of sperm donation, more knowledge is needed on the variety of experiences and attitudes of former donors.

Investigating the long-term impact of having been a sperm donor is particularly relevant since the donor's attitude towards donation may change over time in response to developments in their individual situation and in society as a whole. Changes in personal circumstances, such as having children of their own and/or current partner's attitude towards donation, may prompt a rethinking of previous actions and decisions, or even regret [7]. Furthermore, historical developments in social and political contexts may also change the meaning and impact of sperm donation. For example, the recent commercialization of genetics and the proliferation of direct-to-consumer DNA databases allow donors and donor conceived persons (DCPs) to search for genetic relatives [3,8], and may also inadvertently lead to discovery of donor conception [9]. Thus, donor anonymity could be threatened by technological developments beyond the control of sperm banks and individual donors [10]. Additionally, over recent years, there has been increasing attention to the rights of DCPs to have access to information about donors, and several countries have changed their jurisdiction (some with retrospective effect) regarding donor rights and anonymity [11].

The aim of this study was to explore the experiences and attitudes of men who were sperm donors more than 10 years ago.

## Materials and methods

The study was conducted with an explorative design based on semi-structured, qualitative interviews [12].

### Study context

The study was conducted in the context of Danish legislation on treatment of assisted reproduction, that up until 2007 only allowed anonymous sperm donation. However, since 2007, sperm donors have had the option of anonymous or non-anonymous donation. In **anonymous** donation, the identity of the donor is not known by the recipient or the offspring and

the offspring will not be able to receive any further identifiable information from the sperm bank. In **non-anonymous** donation, the identity of the donor is not known by the recipient or the offspring, however, the offspring will be able to receive the name and latest address on the donor by request to the sperm bank after they reach 18 years of age. Neither anonymous nor non-anonymous donors are able to receive identifying information regarding the DCPs.

## Recruitment

Participants were recruited via Cryos International sperm bank at four branches across Denmark (www.cryosinternational.com). From February to November 2021, former donors were contacted in batch by the Cryos Research Team to be invited to participate in a range of studies, including this one. The inclusion criteria were: 1) > 10 years since last donation, and 2) available contact information. Approximately 200 former donors met these criteria, and in total 77 responded to contact. Recruitment for the present study took place from May to September 2021, and all former donors who responded to Cryos Research Team contact during that period were consecutively invited to participate in the interview study. As a result, a total of 39 former donors were invited to participate and 29 consented to being contacted by the first and second author, who were primary researchers in the study and not affiliated with the sperm bank. They provided potential participants with additional information and, upon consent, an interview appointment was arranged. Out of 29 potential participants, two declined participation, and four could not be reached, thus leaving a sample of 23 former donors.

## Data collection

All interviews were conducted by phone (n = 21) or video (n = 2) and lasted on average 24 minutes (range: 9 to 47 minutes). All interviews were conducted by SL and SB, who are both experienced in qualitative interviewing. The interviews were guided by a semi-structured interview guide informed by current qualitative literature and the interdisciplinary team of authors. Topics in the interview guide and examples of questions can be found in Table 1. The same guide was used for all interviews, but the extent of probing to different topics varied between participants. Before all interviews, oral consent was obtained and recorded, and all participants were encouraged to speak freely and introduce topics that they found relevant. All interviews were audio-recorded and transcribed verbatim. Transcripts were not sent back for checking.

## Data analysis

The material was analysed using reflexive thematic analysis [13,14]. All transcripts were thoroughly read by SL, SB, and a research assistant, TP, who individually developed a set of preliminary codes. These codes were compared, collated and divided into five clusters consisting of three to six codes each related to: 'Reasons', 'Experiences', 'Anonymity', 'Relations' and 'Disclosure'. Three interviews were test-coded independently by SL, SB, and TP for coding reliability, and discrepancies in coding were discussed and settled. Finally, all interviews were coded by TP using NVivo 12 software (QSR International, Melbourne, Australia). The coded material was subsequently read, re-assessed and sorted into themes [13]. For example, the subtheme 'Defining (non-) relatedness' included codes from both 'Experience', 'Relations' and 'Disclosure' clusters. During these analytical, back-and forth processes, an overall theme was identified that captured a recurrent pattern across the data material: Unproblematic and in the past. Four subthemes that represented different aspects of the overall theme were developed: (i) Being a donor was convenient and meaningful; (ii) It's in the past; (iii) Defining (non-)relatedness; and (iv) Thinking about potential contact. Theme and subthemes were then investigated

**Table 1. Examples of topics and questions in the interview guide.**

| Topic | Examples of questions |
|---|---|
| Attitude towards donation | I would like to hear a bit about what it means to you today that you were once a sperm donor? |
| | How often do you think about it? |
| | Has your attitude towards donation and your role as a sperm donor changed over the years? |
| | Have you ever regretted being a donor? |
| | Have you needed any information or support since you stopped? |
| Thoughts on donor conceived persons and anonymity | Can you tell me a bit about the pros and cons of your donor status (anonymous or non-anonymous)? |
| | Looking back, would you choose another donor status? |
| | Do you ever think about the children who came into the world thanks to your donation? (when, why) |
| | How would to describe your relation–if any–to them? |
| | Do you think of these persons as 'your children'? |
| | Have you been contacted by any of them? |
| | Would you be interested in contact with them, why / why not? |
| | How do you feel about donor conceived people that look for their donor? |
| Social network and disclosure of donor status | Are you generally open about being a former sperm donor? |
| | Do your family and friends know that you were a donor? |
| | Do your current partner about it? What is their attitude? |
| | We know, that some donor conceived people search for siblings and maybe donors via online communities and genetic platforms. Do you have any experience with that? |
| | Are you concerned that your anonymity will be circumvented? |

*The full interview guide (in Danish) can be obtained from the first author on reasonable request.

in relation to the full dataset while looking for discrepancies and disconfirming evidence, before writing up the findings. To secure anonymity for participants, all identifying information have been left out of the manuscript and all participants have been assigned a pseudonym.

## Ethical approval

The study was presented to the Central Denmark Region Committee on Health Research Ethics (J. No. 172/2022). According to Danish legislation, research using questionnaires and interviews that do not involve human biological material (§14(2) of the Committee Act) interview studies are exempt from approval from the Committee on Health Research Ethics (https://en.nvk.dk/how-to-notify/what-to-notify). The study was approved by the Danish Data Protection Agency (J. No. 1-16-02-201-21, April 24 2021). Informed oral consent was obtained and recorded prior to all interviews.

## Results

The participating men represented a diverse group in terms of age, profession, and marital status (see Table 2). Prior to and during interviews, many men expressed surprise that former donor's experiences was a topic for research and often added that they were not sure that they would be able to contribute much. By participant choice, many telephone interviews were conducted while the participant was driving home from work, having their afternoon coffee, or walking to their next destination. The overall finding of the analysis was that for most of the

**Table 2. Characteristics of sample (N = 23).**

| Characteristics | |
|---|---|
| **Age** | |
| At interview (years, mean (range)) | 40.9 (33–52) |
| At first donation (years, mean (range)) | 27.0 (19–39) |
| **Time since first donation**\* (years, mean (range)) | 14.8 (12–25) |
| **Donor status** | |
| Anonymous (*n*) | 21 |
| Non-anonymous (*n*) | 2 |
| **Civil status** | |
| Partner (*n*) | 17 |
| Single (*n*) | 6 |
| **Own children** | |
| 0 children (*n*) | 7 |
| 1 child (*n*) | 3 |
| 2 children (*n*) | 8 |
| 3 children (*n*) | 5 |

\*At time of interview.

participants, their former time as a sperm donor was unproblematic, in the past and not a subject for further reflection. This overall finding was reflected in four subthemes describing the meaning of past donorship: (i) Being a donor was convenient and meaningful; (ii) It's in the past; (iii) Defining (non-)relatedness; and (iv) Thinking about potential contact.

## Being a donor was convenient and meaningful

This subtheme contains the men's motivation for becoming a donor, their interaction with the sperm bank at that time, and their reasons for stopping as a donor. The analysis revealed several concurrent motivations for initially applying to be a sperm donor. First, many of the participants had been students at the time of recruitment and considered sperm donation to be an easy way to make extra money. Being a sperm donor was '*easier and funnier*' than other student jobs. Also, several participants described living close to the sperm bank, thus making donation easy:

> *It was mainly the financial part that appealed. And the convenience–pop in and deliver, go home, and collect. And not spend a thought more on it. (Brandon)*

Second, all men mentioned the altruistic opportunity to help people struggling with infertility. Thus, being a sperm donor was considered by them as a win-win situation where monetary and altruistic purposes could be conveniently and meaningfully combined. For three men, however, altruism was the main motivation. They had all experienced the distress of infertility first-hand (e.g., among family or friends):

> *Well, I witnessed how my sister and brother-in-law struggled to have children. I saw first-hand how miserable they were. And when [sperm bank] opened in [city of residence at the time] then I just thought. . . this is a way where I can maybe make a difference for someone. (Chris)*

An additional motivation for some men was curiosity about their sperm quality. Being approved by the sperm bank and having acceptable sperm quality was a source of satisfaction, or even pride, that also provided peace of mind regarding own future fertility:

> *I was motivated by several factors, but honestly, I was curious about sperm quality... if it was good enough. I guess it was a bit of a vanity thing... (Daniel)*

Finally, a main motivation and (for many) the very foundation for even thinking about becoming a donor was the opportunity to be anonymous. Many stressed that the option of anonymity and their confidence in the sperm bank's ability to uphold that anonymity was essential for their decision to become donors:

> *For me it was essential that it was anonymous. I imagine that very few would donate if you couldn't do it anonymously. (Joe)*

> *Only by being anonymous can you put it away again and say: I am donating something, ok, and I will never know any more about it. Or risk being confronted with it in the future. (Kevin)*

Anonymity was often described as a way to protect their future self from unwanted and unpredictable consequences. Two men had chosen to be non-anonymous donors, as they thought that offspring should have right to contact if interested. One of them had been motivated by the possibility of reproducing his genes:

> *I think it's exciting to know that future generations carry my genes. I think it's something deep in our biology, that all humans want to multiply. (Michael)*

Generally, the men had felt well-informed by the sperm bank during their time as a donor; and, in hindsight, had not required any further information or support during or after it finished. The reasons for stopping being a donor varied. A few had their contract cancelled by the sperm bank and a few were unhappy with changes in remuneration, but for the majority it was simply no longer convenient (e.g., due to relocating to a new area and/or starting a family). At the time of the interview, only one participant expressed regret for having been a donor. He had been informed that a diseased child had been born from his donation and expressed feelings of guilt towards the recipients and child. However, today, most men still felt positive about, or even proud of, having been a sperm donor.

## It's in the past

During interviews, most men (including one non-anonymous donor) reported they hardly ever thought about having been a sperm donor. It was described as a *'closed chapter'* that belonged to the past; not a secret, but simply not relevant to their current lives. For these men, many of the interview questions guide were answered with *'I haven't thought about it'* or *'I don't know'*, but many tried to offer more explanation. However, it was obvious that having been a sperm donor did not impact their current lives. Many had now settled in family life and considered donating part of their youth:

> *I still feel good about it [having been a sperm donor]. But I am also somehow happy that it's history. Because now I have my family, I have a life that is quite different from back then... when I was the young bachelor type. (Matt)*

Thus, donating sperm was understood as a temporally bounded event that belonged to the past. For a few men, however, the boundary was less clear and their time as a donor was still present in their current lives. For example, one non-anonymous donor often thought about and looked forward to being contacted by offspring, while an anonymous donor had actively searched for recipients and offspring. Except for the latter, all men were content with their original choice of donor status (non-anonymous or anonymous) and found it important to uphold their contractual rights and obligations towards the sperm bank, the recipients and donor offspring.

## Defining (non-)relatedness

When addressing their perceived, potential relationship or relatedness to donor offspring, many men found it difficult to articulate or define, often asking the researcher to repeat or re-phrase the questions. Most men simply maintained that there was no relationship at all:

> *There is no relationship. At all. That was essential for me going into this. . . I mean, of course there is the 'good cause' and all, but it wasn't my primary motivation and I have no relation to. . . It was a job. I thought of it as a job and that is all [laughs].* (James)

> *It's like an organ or. . . I guess it's a bit cynical, but. . .. [. . .], It's something you pay for, when you want to have a child. I give something, they get it, that's all there is to it really.* (Jacob)

However, when explicitly asked about relatedness, many men acknowledged that they potentially shared genes with DCPs *'out in the world'*, and some would joke about how *'lucky'* or *'poor'* this donor offspring was. Most men made the distinction between genetic origin and fatherhood, and the latter only extended to their relationship to their *'own children'*. It was emphasised that fatherhood is a status and a privilege that is socially and emotionally achieved rather than genetically determined:

> *It's not just a matter of genes. It's raising, protecting, and caring for a child that defines parenthood to me. Not the genes. And I passed the 'raising, protecting and caring' part on to those who received the donation.* (Chris)

To claim the status of fatherhood thus entailed protecting, bringing up and being a part of the children's lives. For the fathers in the sample, all concern and loyalty were firmly directed at their own children, and they stressed that they in no way should or could be perceived as having a father-like relatedness to offspring.

At the time of the interview, sixteen men had children of their own. For some, fatherhood had led them to consider the wellbeing of potential offspring; but they also emphasized how such knowledge could be *'damaging'*:

> *When I was younger, I didn't care [about the fate of offspring]. But today, I have children of my own and I realise how dependent they are–on love and good surroundings. So, in some ways I am concerned [for DCP], and there is a sort of relief that I don't know who they are. [. . .] If one of the children were born into a deeply dysfunctional family, then that would make me upset. I would feel responsible. Therefore, it's better not to know.* (Alex)

The men (fathers and non-fathers) expressed how knowing about their donor offspring and recipients could create unwanted feelings of relatedness and responsibility towards them. This was something that they wanted to protect themselves and their families from. Most men

had informed their current partner about their past as a donor and generally, the partners were reported to be overall accepting. A few men described their wives as a *'a bit unhappy'* with not having the sole rights to the man's gene pool:

> *Back then it wasn't a problem, but over the years. . .I don't know. After we got our own kids, she started thinking about it differently. . . that I might have a few more out there.* (Tyler)

This could indicate that some partners attached meaning to genetic relatedness and that the past donorship threatened the exclusivity of their genetic relatedness. The potential meaning and importance of genetic relatedness was addressed by several men, who acknowledged that such information might be of importance to DCPs. Many men understood genetics to be an explanatory factor in understanding both appearance and personal traits. While maintaining that the recipients were, and would always be, the DCP's parents, many donors accepted that knowing about genetic inheritance could be important to some DCPs:

> *I can definitely relate to that. Particularly when you're a teenager and figuring things out. Who am I? Where do I come from? You're looking for answers, and I understand why one might be interested in biological origin. And maybe some explanations. . . about appearances, but also reactions and behaviors. I think it's really interesting.* (John)

As such, several men expressed sympathy towards some DCPs' desire to meet a donor but felt that their rights to anonymity and the protection of his family should carry more weight.

## Thinking about potential contact

All men reported to trust the security of their data at the sperm bank but also acknowledged the ever-increasing possibilities for online investigations and searches where DCPs could be successful in identifying siblings and/or the donor. Generally, the donors were not well informed about, or interested in, the presence of online ancestry databases, voluntary registers, or Facebook communities; a finding that resonates well with the donor's lack of interest in contact with potential offspring. However, many men also mentioned that the (unlikely) risk of unwarranted contact from an offspring was an inherent uncertainty in sperm donation:

> *I know that sometimes they manage to find the father, even if he's anonymous. If it happens, it happens. I'll deal with it.* (David)

Many men clearly stated that breach of anonymity would be *'highly problematic'* or *'very unfortunate'*, while others described offspring-initiated contact as a matter of *'crossing that bridge when I get there'*. The men who had been anonymous donors had generally not given offspring contact much thought and had not planned for what to do in that unlikely event:

> *I haven't thought about it. I guess it's one of those things where you can't really know until it happens. It's a difficult question and I can't predict my response.* (Chris)

The main concern regarding breach of anonymity was that offspring contact would create an unwanted disturbance in their current lives and families. Fathers expected to be particularly concerned about the response of and impact on their children, none of whom knew at the time of the interview that their father had been a sperm donor, mainly because they were still quite young:

*I also have a family to take care of and that's probably. . . Like, I think I could probably deal with it OK, but what about the rest of the family? [. . .] It will create an imbalance in the family when they have to deal with it too. And that would be the hardest for me, that it would disturb their everyday life.* (Joshua)

Another concern was that offspring-initiated contact would *'force'* a personal relationship onto the man. Many former donors imagined how being confronted with an *'actual, living person'* would probably make it difficult not to get engaged in that person's life, even if they were initially not interested in it:

*I wanted to be anonymous, because. . . well, if someone came around, I just think. . . I think I'm this loving person and I could really just end up seeing them as family. And they already have a family, I know, but I think I couldn't help think that we were family too. That's why I chose to be anonymous.* (Joe)

Several men brought up, that if a donor offspring had gone through such troubles to circumvent anonymity, then they must be very much in need of answers and the men expected to feel morally obliged to respond. While some men imagined that setting boundaries for this relationship or contact could be difficult, others felt that they could maintain very clear boundaries if a donor offspring unexpectedly made contact:

*OK, so let's go have a cup of coffee. Somewhere neutral. And let's talk through it all, could be interesting. And then, that would be the end of it. Like, I am your genetic origin and we can talk about that. But we are not going to discuss me being your dad. 'Cause I'm not.* (Alex)

Such boundaries would allow the man to *'be a decent person'* and respond to a human being who had gone through a lot of trouble while still protecting his family and clearly place them as top priority.

A few men described a fleeting interest in having contact with their donor offspring, mainly due to curiosity about similarities in appearance and personality traits. A couple of men had done a little online research but stopped themselves:

*I am curious about it and I thought that if I could maybe find them [DCP] without making contact. . . It seemed very easy. But then, I guess, knowing about them would create an imbalance. I'd risk my everyday life just to satisfy a curiosity. Not worth it.* (Justin)

Two men, both without children, explicitly expressed interest in meeting their offspring due to curiosity about their lives, their appearance and personal traits. One of them, an anonymous donor, described sometimes *'romanticizing'* the potential relationship with donor offspring (e.g., imagining going on fishing trips). The other, a non-anonymous donor, were excited about the future, potential relationship and imagined himself being like *'an uncle'*. Even so, the men were mindful of the potential vulnerabilities and risks of contact (e.g., the absence of predefined roles and boundaries). Nevertheless, this uncertainty was outweighed by their desire for meeting their donor offspring.

## Discussion

This study provides insights into the experiences of former donors who generally described their donation as a temporally bounded transaction, that belonged in the past and, for most, had little impact on their current lives. The men generally acknowledged DCPs' interests in

knowing about their genetic origin, but emphasized their non-relatedness to offspring. Most of the men had not thought much about a potential breach of anonymity but were mainly concerned that it would disturb their families and force them into a relationship and feelings of responsibility towards their donor offspring.

Overall, our findings are remarkable in that they provide insights into the—often underrepresented—perspective of anonymous donors who are disinterested in identity release and DCP contact. As noted by Kirkman and colleagues [11], most research of donor perspectives included only donors who were willing to participate in studies (a condition in most research) which may create an ascertainment bias towards the more activist communities [15]. In line with this point, much donor research did focus on donors who displayed positive attitudes towards voluntary registries [16], who were actively searching for DCPs [3,17] and/or who had been approached or established contact with DCPs [5,6]. Thus, these perspectives shape the debates about sperm donors' views on and desire for DCP relatedness and relationships. However, a group of donors remain, whose perspectives receive less attention. Kirkman et al. [11] suggested that some donors do not volunteer for research because they value their privacy and do not want to risk their anonymity. Based on the findings of the present study, we add that some former donors do not feel that they have a story to tell. Most of the men in our study thought of sperm donation as a closed chapter and did not desire future contact with their donor offspring. Thus, many of them questioned how they could contribute or why their perspectives were of interest. Thus, we assume that these men would not have volunteered in an open call for participants, but only agreed to participate because they were already participating in a set of studies at the sperm bank.

Our findings echo other studies on sperm donor motivations to include altruism, the monetary reasons and for some an interest and pleasure in having good quality sperm [18–21]. Most men in this study were recruited as anonymous donors, which was mandatory in Denmark up until 2007. Today, both anonymous and non-anonymous donation is possible. Several studies have indicated that the different recruitment regimes attract different types of men [2,22,23]. A recent survey of 233 active donors in Denmark and the United States compared anonymous and non-anonymous donors [1] and showed that non-anonymous donors were older and were more likely to have a partner. They also thought significantly more about their potential donor offspring and were more likely to want information about their offspring such as number of children, gender, and health status. The donors did not differ in their motivations. However.as found in previous studies, the anonymous donors were significantly more likely to *not* want information about DCPs, which also resonates with the findings in the present study. Interestingly, these differences in characteristics are often discussed with an implicit positive bias towards the older and more settled type of man opting for open donor status. For example, in their investigation of personality characteristics of men who became donors in a system allowing only non-anonymous donors, Sydsjö and colleagues [24] conclude that a positive effect of an open-donor system is that it attracts men who have reasons for donating other than financial motives. This value-based interpretation of donor motivation shows how body commodification (sperm as product) is perceived as less honourable than altruism (sperm as gift). Implicit in these views are also specific assumptions about the relation between donor and donor offspring, because the transaction of a product is without emotional commitment and responsibilities, whereas the exchange of a gift implies potential moral obligations [25]. The men in the present study were very clear that their contribution was delivering a product and that the moral obligations towards the DCP had been delegated solely to the parent recipients.

Our findings demonstrate how the majority of men clearly demarcated the line between genetic contribution and fatherhood, and part of this demarcation was achieved by not

knowing about the donor offspring or recipients. Though some expressed curiosity, our findings point to an understanding of knowledge as correlated with feelings of responsibility and potential relatedness. Something that most of the men were not interested in. Similarly, there may be groups of parent recipients and DCPs that prefer a disinterested donor and who prefer thinking about sperm donation in terms of product and transaction [10]. A donor recruitment regime that allows different types of donors also allows recipients to actively chose an anonymous donor who values privacy and who will most likely not search for recipients or donor offspring in the future. Likewise, recipients in a system that allows choice could be expected to honour the donor's anonymity. As for the DCPs, studies have shown that some choose to search for anonymous donors. However, more knowledge is needed on the perspectives of DCPs who do *not* search for the donor and thus also uphold a contract of anonymity and no contact. Like the former donors in the present study, these DCPs may not come forth in research exactly because they value their privacy and do not feel that they have a story to tell.

Only one man had regrets about having been a donor. The birth of a child with birth defects caused him to feel responsible for the bad fate of the child and the recipients. It is unknown if the child's condition was indeed heritable or related to the donation; however, more research is needed on donors' responses when a diseased child or a heritable condition is discovered in DCPs.

Almost all donors in the present study had informed their partner about their past donation. This is very high given that other studies have shown that many donors did not tell or involve their partner [2]. An important reason for this was the donor's concern about negative responses from the partner [6]. The legitimacy of this concern was supported by a recent study showing that 28% of women would find it difficult to cope with the past donation of their partner [26]. It must be kept in mind, that our study reports on the donor's perceptions of their partner's feelings and attitudes and may be biased by the donor's more carefree approach. Some donors hinted at the fact that their partner was not happy about past donations mainly due to not having exclusive rights to the donor genes and/or the risk of future offspring contact. As such, our findings resonate with other qualitative studies showing the partner's concerns regarding obligations linked to the donation that may affect their family in the future [27,28]. The donors in our study had not told their own children because of their young age but expressed concerns about how their own children would react when they were told. This worry has also been found in other studies [28].

Interviewing about perspectives that are largely silent and unimportant in an informant's everyday life, gave rise to some ethical considerations. During interviews, the interviewers were attentive to the men's right not to know and careful about introducing or imposing concerns. For example, the questions regarding DNA databases were asked in more general terms and only further prompted if the participant expressed some knowledge or interest. Interviews made it obvious that for many men, sperm donation had not been a cause for much contemplation in recent years and several participants developed their thoughts and attitudes during the interview. Therefore, we were observant of potentially normative or biased language in the interview guide and emphasized the legitimacy of all types of attitudes and experiences.

The main strength of the current study is the sample selection that encouraged anonymous donors to come forth and that data collection continued until adequate information power was estimated to have been met. A limitation is that the sample is not necessarily representative of Danish sperm donors and the findings are thus not generalizable in the quantitative sense. More knowledge is needed on the experiences and attitudes of donors with extended profiles and non-anonymous donors, as well as research on former donors' experiences in cases where anonymity is breached. However, these qualitative findings present a range of attitudes and

experiences among former donors that may be transferable to similar contexts where donor anonymity is an option.

In conclusion, this study reports on former donors who might not have volunteered for research due to lack of interest or protection of privacy. The findings illuminate the perspectives of former donors who value anonymity and who are not interested in knowledge about donor conceived persons or parent recipients.

## Acknowledgments

The authors also wish to thank Emilie Møller Lassen and other staff at Cryos International, who assisted with recruitment of potential participants. Thanks to research assistant Tina Petersen for valuable assistance with coding and analysis. And most of all, thank you so much to all the participating men who took time to share their experiences with us.

## Author Contributions

**Conceptualization:** Stina Lou, Stine Willum Adrian, Allan Pacey, Guido Pennings, Ida Vogel, Anne-Bine Skytte.

**Data curation:** Stina Lou, Stina Bollerup, Anne-Bine Skytte.

**Formal analysis:** Stina Lou, Stina Bollerup, Morten Deleuran Terkildsen, Ida Vogel, Anne-Bine Skytte.

**Funding acquisition:** Ida Vogel, Anne-Bine Skytte.

**Project administration:** Stina Lou, Stina Bollerup.

**Supervision:** Morten Deleuran Terkildsen, Stine Willum Adrian, Allan Pacey, Guido Pennings, Ida Vogel, Anne-Bine Skytte.

**Validation:** Morten Deleuran Terkildsen, Stine Willum Adrian, Allan Pacey, Guido Pennings, Ida Vogel, Anne-Bine Skytte.

**Writing – original draft:** Stina Lou, Stina Bollerup.

**Writing – review & editing:** Morten Deleuran Terkildsen, Stine Willum Adrian, Allan Pacey, Guido Pennings, Ida Vogel, Anne-Bine Skytte.

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
