## [Decision Letter · Decision Letter 0]

14 Nov 2022

PONE-D-22-21878Experiences and attitudes of Danish men who were sperm donors more than 10 years ago; a qualitative interview studyPLOS ONE

Dear Dr. Lou,

Thank you for submitting your manuscript to PLOS ONE. After careful consideration, we feel that it has merit but does not fully meet PLOS ONE’s publication criteria as it currently stands. Therefore, we invite you to submit a revised version of the manuscript that addresses the points raised during the review process.

Dear Stina Lou

About your paper entitled “Experiences and attitudes of Danish men who were sperm donors more than 10 years ago; a qualitative interview study” We have considered that the paper is interesting and could potentially be published in a new versión (the decisión is Major Revision) that takes into account the observations made by the referees.

I am attaching the referee's comments, which will help to explain the reasons for our decision. As you can see, the reviewer finds the paper to be of interest, but raises a number of significant concerns. I hope the reports may be useful if you are considering revising the paper for re-submission to Plos One.

Yours sincerely, Sónia Brito-Costa

We look forward to receiving your revised manuscript.

Kind regards,

Sónia Brito-Costa, Ph.D.

Academic Editor

PLOS ONE

Journal Requirements:

“This study was funded by Cryos International and the Novo Nordisk Foundation (Grant No. NNF16OC18722).”

“This study was funded by Cryos International, https://www.cryosinternational.com/da-dk/dk-shop/professionel/forskning/vores-videnskabelige-forskning/ (to SL) ,and the Novo Nordisk Foundation, https://novonordiskfonden.dk/en/ (Grant No. NNF16OC18722 to IV). Cryos International assisted with recruitment of participants, but otherwise the funders had no role in study design, data collection and analysis, decision to publish, or preparation of the manuscript.”

“SL, SB, MDT and IV declare no conflict of interest. AP, GP and SWA are members of the Cryos External Scientific Advisory Committee. GP has received honorarium for this. ABS is an employee of Cryos International.”

Reviewers' comments:

Reviewer's Responses to Questions

**Comments to the Author**

1. Is the manuscript technically sound, and do the data support the conclusions?

Reviewer #1: Yes

Reviewer #2: Yes

2. Has the statistical analysis been performed appropriately and rigorously? 

Reviewer #1: Yes

Reviewer #2: N/A

3. Have the authors made all data underlying the findings in their manuscript fully available?

Reviewer #1: Yes

Reviewer #2: Yes

4. Is the manuscript presented in an intelligible fashion and written in standard English?

Reviewer #1: Yes

Reviewer #2: Yes

5. Review Comments to the Author

Reviewer #1: Very interesting and important longterm follow up of sperm donors.

The manuscript is well written and I have only a few concerns.

My first concern is about how the donors who were possible to interview were selected (39 donors)? Please develop that part ie how many in total were available >10 years, if not all there is a need for mor information regarding the criteria for being recruited.

My second concern is about the semi structured questions. How were they used? Were all these questions asked to all of the participants?

It would improve the understanding if you describe a little more (few examples) how your process of doing reflexive thematic analysis was performed/developed regarding from the main text from the interviews?

The analysis was based on interviews 9-47 minutes (mean 24). 9 minutes are very short to get substance for an analysis. You base your results on 21 anonymous and separate 2 non-anonymous donors answers. Why did you include only 2 non-anonymous, or why did you include these two at all? The main message may be more clear if focus and discussion only were about anonymous donors. As you write (and participants answers show) the non-anonymous donors think differently.

Interesting discussion that add insight about donors longterm perspectives.

Reviewer #2: This article is relevant to a better understanding and characterization of gamete donors. Most studies focus on donation motivations and are carried out before or shortly after donation, not allowing this long-term analysis.

Regarding the submitted article, there are 3 fundamental points that I want to emphasize:

1. The abstract should be improved, as it is not particularly appealing to the potential reader. I also emphasize the need to correct the second sentence of its "methods" section and improve the conclusions;

2. Although in Denmark this type of study does not require the statement of an Ethics Committee, this statement is required by PLOS One "if the study involved human participants", so the authors, if they intended to submit their study to this journal, should have read the submission rules and prepare theirr project accordingly. The decision of whether or not to accept the publication without fulfilling this requirement will, of course, be at the discretion of the Editor;

3. Despite being a qualitative study, I believe that the designation in the results, repeatedly, of terms such as "many", "for the majority", "a few", "several" and "most" should be replaced by the number of participants that we are actually mentioning in each case.

6. PLOS authors have the option to publish the peer review history of their article (what does this mean?). If published, this will include your full peer review and any attached files.

Reviewer #1: No

Reviewer #2: No

---

## [Author Response · Author response to Decision Letter 0]

4 Jan 2023

Response to decision letter

Dear Sónia Brito-Costa, Academic Editor, PLOS One

First, we wish to thank the reviewers for insightful and very constructive comments on our manuscript. We have revised the paper in accordance with the reviewers' suggestions and believe it has improved the manuscript.

Below we address the comments and explain how we have aimed to accommodate them in the revision of the manuscript. All reference to line numbers are in the track changes version of the revised manuscript.

We hope that the revision meets your expectations and look forward to hearing from you.

Best regards,

Stina Lou

Journal requirements 

1. Please ensure that your manuscript meets PLOS ONE's style requirements. 

 - We have revised the manuscript in accordance with PLOS ONE's formatting requirements, including title page, tables and manuscript headings (PLOSOne_formatting_sample_main_body.pdf (storage.googleapis.com)

2. Please remove any funding-related text from the manuscript and let us know how you would like to update your Funding Statement. Please include your amended statements within your cover letter; we will change the online submission form on your behalf.

- Updated Funding Statement: This study was funded by Cryos International (personal grant to SL, no grant number assigned) and the Novo Nordisk Foundation (Grant No. NNF16OC18722 to IV). Staff at Cryos International assisted in recruitment of participants for the study, but the funders had no role in the data collection, analysis and interpretation of results.

“SL, SB, MDT and IV declare no conflict of interest. AP, GP and SWA are members of the Cryos External Scientific Advisory Committee. GP has received honorarium for this. ABS is an employee of Cryos International.” Please confirm that this does not alter your adherence to all PLOS ONE policies on sharing data and materials, by including the following statement: "This does not alter our adherence to PLOS ONE policies on sharing data and materials.” (as detailed online in our guide for authors http://journals.plos.org/plosone/s/competing-interests). If there are restrictions on sharing of data and/or materials, please state these. Please note that we cannot proceed with consideration of your article until this information has been declared.

 - Updated Competing Interests statement: SL, SB, MDT and IV declare no conflict of interest. AP, GP and SWA are members of the Cryos External Scientific Advisory Committee. GP has received honorarium for this. ABS is an employee of Cryos International. This does not alter our adherence to PLOS ONE policies on sharing data and materials. Due to anonymity issues, data material can only be made available upon reasonable request to the first author.

 - Updated Data Availability Statement: For this study, participants only consented to external data sharing in anonymized form. Since full transcripts cannot be fully anonymized due to the highly individual context, the transcripts can only be made available upon reasonable request and special conditions may apply. Any requests concerning data access can be directed to stina.lou@rm.dk

Reviewer #1 

1. Very interesting and important longterm follow up of sperm donors.

The manuscript is well written and I have only a few concerns.

 - Thank you for this positive feedback!

2. My first concern is about how the donors who were possible to interview were selected (39 donors)? Please develop that part ie how many in total were available >10 years, if not all there is a need for mor information regarding the criteria for being recruited. 

 - Good point. We have added more information on the recruitment process, L121-28

3. My second concern is about the semi structured questions. How were they used? Were all these questions asked to all of the participants?

 - The same guide was used for all interviews, but the extent of probing to different topics varied between respondents. We have added this information to the manuscript, L140-41

4. It would improve the understanding if you describe a little more (few examples) how your process of doing reflexive thematic analysis was performed/developed regarding from the main text from the interviews? 

 - Yes, transparency is important! We have now revised the section and added more examples to our process of doing reflective thematic analysis, L154-67 

The analysis was based on interviews 9-47 minutes (mean 24). 9 minutes are very short to get substance for an analysis. We completely agree. And that was also an argument for continuing recruitment until we had a relatively large sample to ensure sufficient substance in the analysis. For some of these men, having been a donor was completely unproblematic and not something they were used to articulating (which is an interesting finding in itself). Therefore, with a few men, the interviews were short, because they mainly provided one-syllable answers and/or did not respond to/recognize some of the topics and potential problems, that we asked about. Luckily most interviews were longer and with more substance. 

5. You base your results on 21 anonymous and separate 2 non-anonymous donors answers. Why did you include only 2 non-anonymous, or why did you include these two at all? The main message may be more clear if focus and discussion only were about anonymous donors. As you write (and participants answers show) the non-anonymous donors think differently. 

 - Reviewer 1 touches an important point that we have also discussed in the research team several times. However, we decided to 'keep' the non-anonymous donors, because they provide an interesting juxtaposition to our main findings. 

5. Interesting discussion that add insight about donors long-term perspectives. 

 - Thank you!

Reviewer #2 

1. The abstract should be improved, as it is not particularly appealing to the potential reader. I also emphasize the need to correct the second sentence of its "methods" section and improve the conclusions; 

 - Sorry about that second sentence! We have thoroughly revised the abstract to make it more appealing, L34-71.

2. Although in Denmark this type of study does not require the statement of an Ethics Committee, this statement is required by PLOS One "if the study involved human participants", so the authors, if they intended to submit their study to this journal, should have read the submission rules and prepare theirr project accordingly. The decision of whether or not to accept the publication without fulfilling this requirement will, of course, be at the discretion of the Editor; 

 - We have uploaded an ethics statement with the submission. Also, we have revised the ethics statement to clarify that the project has been presented to the ethics committee, L173-75

3. Despite being a qualitative study, I believe that the designation in the results, repeatedly, of terms such as "many", "for the majority", "a few", "several" and "most" should be replaced by the number of participants that we are actually mentioning in each case. 

 - There are very divergent attitudes towards the use of numbers in qualitative research. We believe that numerical values can be misleading in qualitative research, particularly if they are read as 'evidence' for a particular finding. A qualitative analysis should describe a pattern or an overall theme. Within this analysis, data points that appear less frequently may also have great value, for example to shed light on varied points of view. We hope that reviewer #2 can accept this explanation for why we have chosen not to add specific numbers to the manuscript

---

## [Editor Report · Decision Letter 1]

13 Jan 2023

Experiences and attitudes of Danish men who were sperm donors more than 10 years ago; a qualitative interview study

PONE-D-22-21878R1

Dear Dr. Lou,

We’re pleased to inform you that your manuscript has been judged scientifically suitable for publication and will be formally accepted for publication once it meets all outstanding technical requirements.

Kind regards,

Sónia Brito-Costa, Ph.D.

Academic Editor

PLOS ONE
---

## [Editor Report · Acceptance letter]

19 Jan 2023

PONE-D-22-21878R1 

Experiences and attitudes of Danish men who were sperm donors more than 10 years ago; a qualitative interview study 

Dear Dr. Lou:

I'm pleased to inform you that your manuscript has been deemed suitable for publication in PLOS ONE. Congratulations! Your manuscript is now with our production department. 

Kind regards, 

on behalf of

Dr. Sónia Brito-Costa 

Academic Editor

PLOS ONE